# Experience of establishing and coordinating a nationwide network for bidirectional intussusception surveillance in India: lessons for multisite research studies

The INCLEN Intussusception Surveillance Network Study Group

**Correspondence to**
Manoja Kumar Das;
manoj@inclentrust.org

## ABSTRACT

**Objectives** To document and share the process of establishing the nationally representative multisite surveillance network for intussusception in India, coordination, data management and lessons learnt from the implementation.

**Design** This study combined both retrospective and prospective surveillance approaches.

**Setting** 19 tertiary care institutions were selected in India considering the geographic representation and public and private mix

**Participants** All children under-2 years of age with intussusception

**Primary and secondary outcome measures** The experience of site selection, regulatory approvals, data collection, quality assurance and network coordination were documented.

**Results** The site selection process involved systematic and objective four steps including shortlisting of potential institutions, information seeking and telephonic interaction, site visits and site selection using objective criteria. Out of over 400 hospitals screened across India, 40 potential institutions were shortlisted and information was sought by questionnaire and interaction with investigators. Out of these, 25 institutes were visited and 19 sites were finally selected to participate in the study. The multistep selection process allowed filtering and identification of sites with adequate capacity and motivated investigators. The retrospective surveillance documented 1588 cases (range: 14–652 cases/site) and prospective surveillance recruited 621 cases (range: 5–191 cases/site). The multilayer quality assurance measures monitored and ensured protocol adherence, complete record retrieval and data completeness. The key challenges experienced included time taken for obtaining regulatory and ethical approvals, which delayed completion of the study. Ten sites continued with another multisite vaccine safety surveillance study.

**Conclusion** The experience and results of this systematic and objective site selection method in India are promising. The systematic multistep site selection and data quality assurance methods presented here are feasible and practical. The lessons from the establishment and coordination of this surveillance network can be useful in planning, selecting the sites and conducting multisite and surveillance studies in India and developing countries.

### Strengths and limitations of this study

► Provides a systematic study site selection process using objective methodology.

► Documents the experience with prospective and retrospective surveillance data collection with multilayered data quality assurance process.

► The study site selection and quality assurance measures may serve as reference for suitable adaptation in different research topics and contexts.

► The lessons are applicable only to the multisite studies.

► The documentation of contractual and financial management challenges may be limited.

## INTRODUCTION

Intussusception is an acute emergency in children and most commonly occurs in infants aged 4–10 months.[1 2] Although some are transient and resolve spontaneously, if not intervened timely, it may lead to bowel ischaemia and perforation and may even be fatal.[1] Intussusception has been reported as an adverse reaction with rotavirus vaccines (RVV) with variable risks ranging from no increased risk to low risk (1–2 additional cases per 100 000 vaccinated infants) across different countries with different RVVs.[3–12] The intussusception background rate varies widely across different countries, 9 (Bangladesh) to 328 (South Korea) per 100 000 infants.[13] Limited information from India reported its incidence between 17.7 and 254 cases per 100 000 child years.[14 15] Several other studies were single or few centre studies and varied in case definitions, age groups, reference periods and methodology.[12–14]

In view of the vaccine safety concern and limited information on intussusception in India, the National Technical Advisory Group (TAG) on Immunisation recommended

vaccine safety surveillance along with RVV introduction.[16] This surveillance network aimed to generate background information on intussusception epidemiology Indian children to inform the policy and programme related to RVV introduction and serve as baseline for future surveillance to identify any change after vaccine introduction and address the vaccine safety concerns. The objectives were to: (1) establish a surveillance network of public and private hospitals ensuring regional representation and data capturing system; (2) undertake retrospective surveillance to document the intussusception epidemiology over past 5 years; (3) undertake prospective surveillance to document the intussusception epidemiology over 18 months and potential linkage with RVVs and (4) build capacity of the investigators and institutions.

Multicentre studies face several challenges related to design, site selection, regulatory approvals, study conduct, coordination, data management and dissemination.[17–20] The lessons from multisite studies in India are limited. The intussusception surveillance study has been successfully completed and the network has evolved over time to undertake more vaccine safety studies. The purpose of this paper is to share the process of network establishment, coordination, data management and lessons learnt from such nationwide network research.

## METHODS

### Study design

This study combined both retrospective and prospective surveillance. The experiences and lessons presented here are based on the concurrent documentation of processes and retrospective review of the study documents.

### Study governance

The Central Coordinating Unit (CCU) constituted investigators, research staffs and administrative and finance team. For technical integrity and implementation monitoring, a TAG was constituted with 17 Indian and international experts in vaccine safety, surveillance, immunisation programme, public health, child health, paediatric surgery, radiology and medical record system, representation from Ministry of Health and Indian Council of Medical Research.

### Site selection

We followed a four-step systematic study site selection process (figure 1). *Step 1 (Shortlisting):* We categorised the states into four regions (north, south, east and west) and screened the tertiary care hospitals (medical colleges and private hospitals) in these regions from the websites. *Step 2 (Screening):* We solicited information about the case load, clinical and diagnostic capacities (paediatrics, paediatric surgery and radiology), medical record system, ethical and administrative approvals from the shortlisted institutions using questionnaire and telephonic interactions (online supplemental document 1). *Step 3 (Evaluation):* A TAG member visited the institutions to assess the institution capacity (clinical facility, case documentation, research support system and medical record-keeping system) and interacted with the potential investigator(s), department and institution leaderships to assess the commitment, institution support and research environment (online supplemental document 2). *Step 4 (Site finalisation):* Based on the questionnaire, interaction with investigators and TAG member feedback, the study sites were finalised.

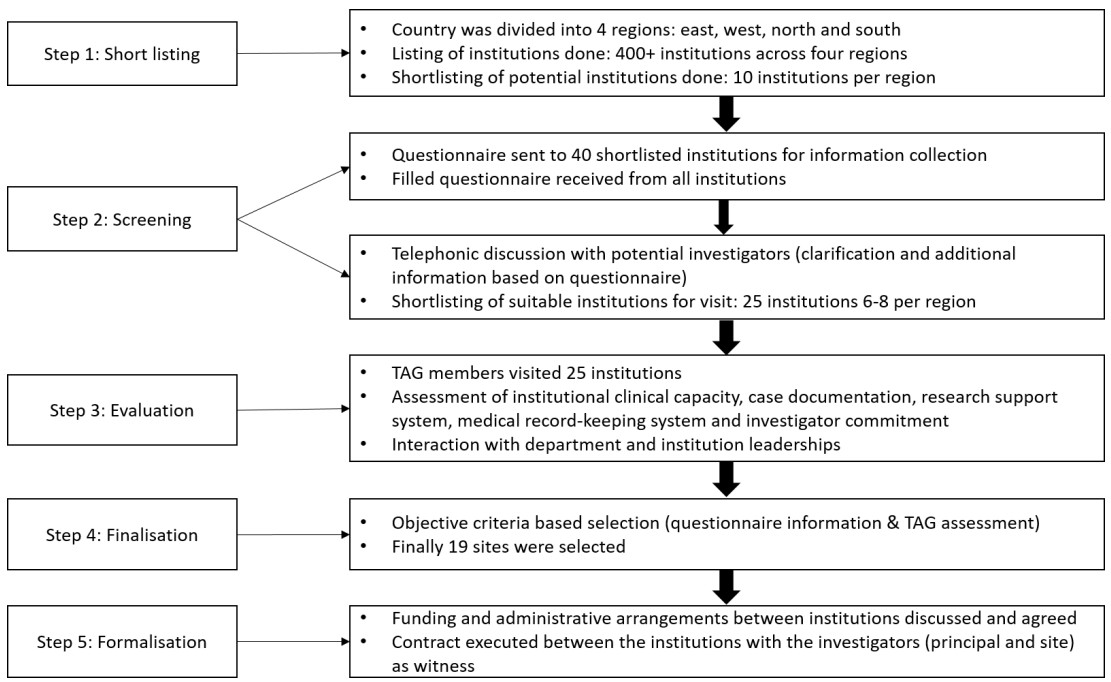

**Figure 1** The steps of the site selection process. TAG, Technical Advisory Group.

## Protocol finalisation and study tool development

In multisite research projects, consensus building among the investigators on the study protocol, case record forms (CRFs), data management and monitoring are critical. The study protocol and CRFs were shared and piloted at the study sites. The protocol, study tools, implementation, data management and monitoring were discussed in detail during protocol finalisation workshop involving lead investigators from all sites and TAG members.

## Regulatory procedures

As this study involved international funding, the approval from Health Ministry Screening Committee (HMSC) was obtained. The study protocol documents were submitted to the participating institutes for ethical approvals. Following the ethical approval, the study was implemented at the sites.

## Study implementation process

Before implementation of the study at the sites, the following four items had to be completed: (1) institute administrative approval; (2) ethical approval; (3) agreement executed with the institution and (4) research staff selected and trained.

## Data collection, management and monitoring

Separate CRFs and log sheets were prepared for the retrospective and prospective data collection. The retrospective data collection involved multiple sources; case records, registers from different departments and operation theatres and medical records section. For case record retrieval, the sites following International Classification of Diseases (ICD, ICD-9 or ICD-10) system, the ICD codes for intussusception or acute abdomen conditions were used (online supplemental document 3) (published in methodology paper).[21] At the other sites, cases were identified as per the diagnoses. For the prospective data collection, all admitted children were screened to identify the suspected cases, who were followed till final diagnosis. The children with intussusception were recruited and data collected. A weekly reporting from the sites was solicited for tracking progress. Initially, a weekly call with the study teams and later fortnightly to monthly calls were done to monitor the progress and address challenges. Based on the data collection progress reports, a monthly bulletin was prepared indicating the progresses and data quality. The completed study tools were sent periodically to CCU. The data received at CCU were reviewed for completeness, correctness and queries were resolved with reference to the source documents. Double data entry was done using customised data entry platform with inbuilt data matching programme. The data were archived in the server with authorised access and regular backup. Data analysis was done by under guidance of TAG. The data collection, flow and management followed for prospective and retrospective surveillance are shown in figures 2 and 3.

## Quality assurance measures

Multiple quality assurance measures focused on protocol adherence and data quality. After initiation of data collection, TAG members visited each study site during the retrospective surveillance to verify: (a) protocol adherence; (b) identification of suitable cases and (c) data abstraction. The corrections/clarifications and repeat training was imparted as per TAG observations. After 6 months, during the prospective surveillance, TAG member again visited to review the: (a) patient screening; (b) tracking and eligible patients identification; (c) obtaining consent; and (d) data abstraction.

## Patient and public involvement

No patients or the public were involved in the design, conduct, reporting and dissemination of research.

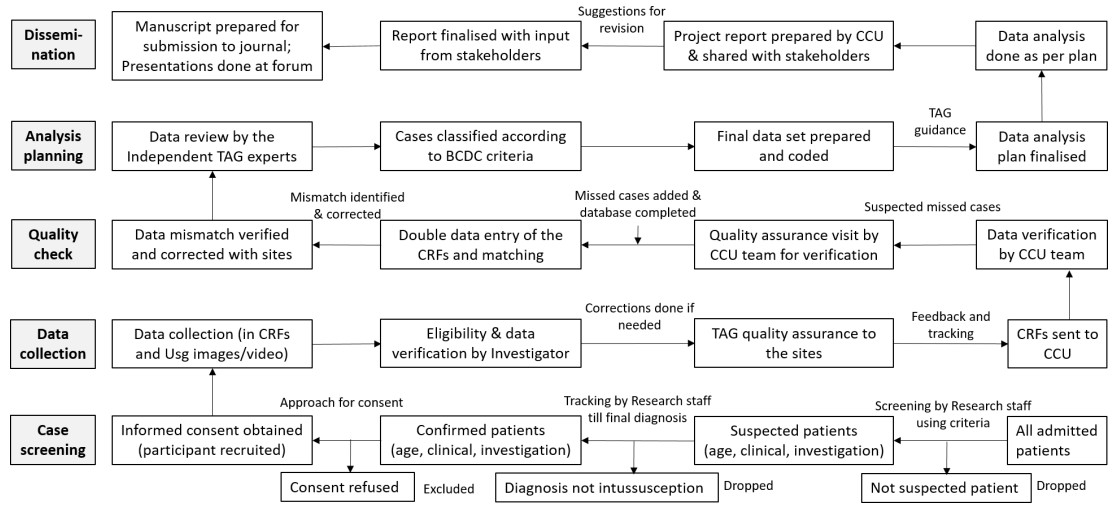

**Figure 2** The data collection and management flow for prospective surveillance. BCDC, Brighton Collaboration diagnostic criteria; CCU, Central Coordinating Unit; CRF, Case record form; TAG, Technical Advisory Group; Usg, Ultrasound.

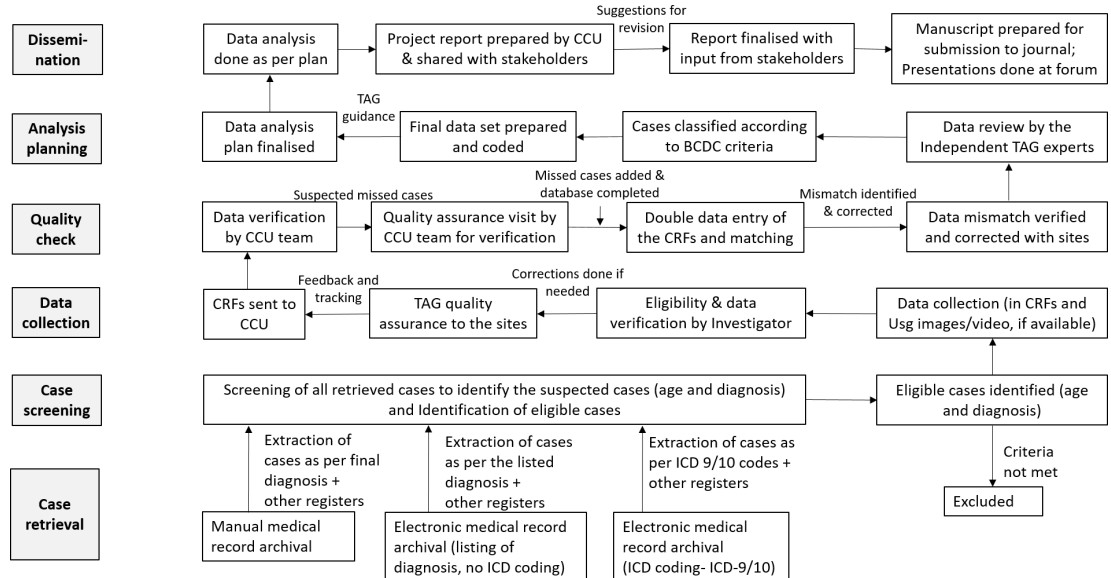

**Figure 3** The data collection and management flow for retrospective surveillance. BCDC, Brighton Collaboration diagnostic criteria; CCU, Central Coordinating Unit; CRF, Case record form; ICD, International Classification of Diseases; TAG, Technical Advisory Group; Usg, Ultrasound.

## RESULTS

### Study timelines

The study was sanctioned in November 2014. The HMSC application was submitted in December 2014 and approved in May 2015. The formal study site selection initiated after the HMSC approval and was completed by August 2015. The protocol finalisation workshop was held in October 2015. Following the ethical approvals, the retrospective surveillance was initiated at majority of the sites in February 2016. The prospective surveillance was initiated in April 2016 and continued through September 2017. The data entry, cleaning and analysis and report drafting were completed by June 2018.

### Study site selection

Out of over 400 hospitals screened across regions, 40 potential hospitals/institutions (10 per region) were shortlisted. While preparing the database, many institutions did not have adequate information on their websites about the facilities and faculty members or specialists. The potential investigators from these 40 shortlisted institutions were invited to submit desired information using a questionnaire via email and all responded. Telephonic discussions was held with the potential investigator(s) from these institutions (45–60 min) for additional information or clarifications. Based on the criteria and consultation with TAG, 25 institutes were visited. Based on the questionnaire, interactions and TAG member assessment, 19 institutions (north region: 5 sites, 3 public and 2 private; south region: 5 sites, 2 public and 3 private; east region: 6 sites, 5 public and 1 private; west region: 3 sites, 2 public and 1 private) were selected (figure 4). The study originally planned for 17 sites. Two more sites from the states where RVV was scheduled for introduction were added later. Several potential investigators requested the CCU team to discuss with their institute leadership for permission and allow him/her to lead the project. The CCU team succeeded in all expect two institutions, which could not be included. The site selection process was delayed by 7 months due to the delay in HMSC approval.

### Regulatory approvals

The ethics approvals from the sites needed average 4 months (range: 1–8 months). No protocol amendment was needed.

### Data collection

During July 2010 and March 2016 (retrospective surveillance period), out of 42 866 admitted under-2 years children, 2092 suspected cases were identified and 1588 confirmed intussusception cases were recruited.[22] The cases recruited at study sites ranged from 14 to 652. While five sites documented <20 cases each, two sites contributed >100 cases each. During April 2016 and September 2017 (prospective surveillance period), out of 6300 hospitalised under-2 years children, 1203 suspected cases were identified and 621 confirmed intussusception cases were recruited.[23] The cases recruited at the study sites ranged from 5 to 191 cases. While seven sites recruited <10 cases each, three sites contributed >50 cases each. At one study site, very few cases were retrieved from the retrospective and prospective surveillance, contrary to the anticipation. The descriptive data analysis was done at pooled and regional levels to document the epidemiology of intussusception. The variables and data parameters between the regions were compared with identify the similarities and differences.

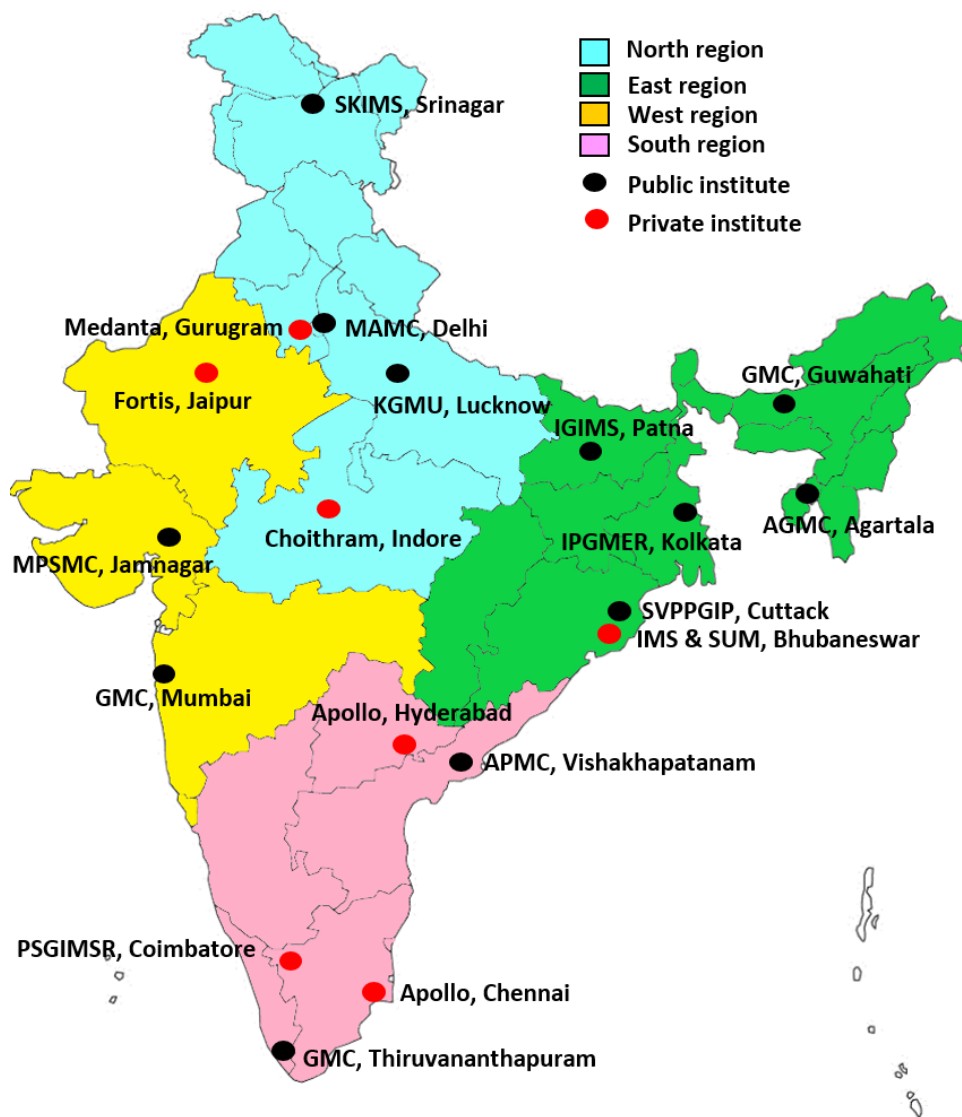

**Figure 4** The geographic distribution of the network study sites. Note: The image was created for the research project by the investigators.

## Medical record retrieval

Twelve (seven private and five public) institutions had electronic medical record (EMR) system. The ICD system was used at all sites except the two private hospitals, where the diagnosis was used for listing and archival. The medical records were accessible at all study sites except one site, where the research staff faced challenge in timely and adequate access to the medical records. The medical records required organisation at this site and thus the retrospective data collection took longer. The record retrieval was quicker at institutions with EMR system and retrospective data collection was completed in 2–4 months. At two sites, the retrospective data collection took longer 6 and 10 months, due to higher case load and challenge in record retrieval, respectively.

## Radiology documentation

The ultrasound digital images and report were available for all prospectively recruited cases. There were variations in the ultrasound finding documentations and details in

the reports, which required clarification from the radiologist for data capturing. Ultrasound reports were available at all but one hospital, where the findings were recorded in the case sheet. Digital records of ultrasounds were stored for minimum 3 months to 5 years. These ultrasound reports, digital films and clinical records were reviewed by independent TAG members for confirmation and classification according to Brighton Collaboration Criteria.[1]

## Vaccination information

Majority of the parents were not carrying the vaccination card at the time of hospitalisation and about half of them came from outside the city. The research staffs pursued to obtain the vaccination card during hospitalisation and after discharge. The vaccination card after discharge was obtained through email, mobile message and also self-addressed and stamped envelopes handed over to the parents. The vaccination information was available for 78.4% of the prospective cases.

## Clinical case management

At five sites only surgery was done. At two sites, reduction was available during daytime only and surgery during other period and at 12 sites, both methods were available always.

## Data quality assurance

TAG members made two quality assurance visits, first during the retrospective surveillance period, second during the prospective surveillance period and provided written feedback. The retrospective surveillance was at risk of missing cases. As it was difficult to compare the absolute case numbers across the centres and periods, we derived intussusception case rate per 1000 paediatric surgery admissions. After completion of 1 year of prospective surveillance, third quality assurance visit was made by the data management team to 11 sites with >30% differences in the case rates between retrospective and prospective periods. The team targeted verifying the confirmed cases, suspected cases and total admissions during 2015, 2016 and 2017 (until visit). The team identified three missed cases (one each at three sites).

## Investigators commitment and transition

Despite the systematic selection process, the involvement of the investigators in the day-to-day operations and data verification was lesser than expected at three sites. At three sites, the lead investigators transitioned without any impact on the study activities.

## Research staff issues

Although one research staff per site was planned, at seven sites, two research staffs were engaged considering the efforts needed for data retrieval and case load. These research staffs were trained through regional training workshops followed by hands-on training. At eight sites, the research staffs changed during the study period, once at six sites and twice at two sites. All the new staffs were trained by the CCU team member through site visit followed by virtual support. The investigators swiftly ensured replacements and no loss of cases. At CCU, there were three rounds of research staff transition.

## Intradepartment and interdepartment coordination

At all institutions the nurses, residents and faculty members supported the surveillance. The nurse and residents at most places informed the research staff about any confirmed case, even on holidays. If the patient was discharged on any holiday, the research staffs attended and collected the data. Intradepartmental and interdepartmental coordination for data collection were smooth at all sites except two sites, where the participation from paediatric department was limited. As most of the intussusception cases were directly admitted to or transferred to paediatric surgery department on diagnosis, it did not affect the case screening and recruitment.

## Financial management

In view of the foreign funding source and Foreign Contribution Regulation Act (FCRA) regulation, the funds could not be transferred to most institutes without FCRA approval. Thus, the fund was managed by the CCU for most of the institutes. Despite no fund transferred to the institutes, agreements were executed between the institutes with the investigators as the witnesses. We experienced challenges with some institutes in explaining the FCRA obligations prior to agreement execution.

## Study tenure

Although planned for 30 months, the study was completed in 43 months. The delays in HMSC and ethics approval at sites were the key reasons for the delay.

## Dissemination

Five peer-reviewed manuscripts have been published and seven manuscripts are under review or in preparation. The manuscripts have all the site investigators as authors, either individually or as group. The findings have been shared with the Ministry of Health and Family Welfare, WHO and other key partners. Over 22 oral presentations had been made at national, regional and international meetings.

## Capacity building and transition

After this study, several sites were included in other studies coordinated by the CCU, 10 sites in one study and four sites in another study. Four site investigators collaborated in other research projects. Several of the research staffs at these sites continued with the new studies.

## DISCUSSION

This multicentre sentinel surveillance successfully generated clinical and sociodemographic epidemiology information on intussusception in children including the regional the variations. The network adopted a systematic site selection process. A common understanding among the investigators was ensured and transmitted to the research staffs for appropriate implementation. The effective coordination, data management and quality assurance measures ensured high-quality data. Despite the efforts, there were challenges related to the regulatory and administrative challenges, which affected the implementation timelines. The continuation of several sites and investigators in other studied demonstrated the strong collaboration and confidence coupled with capacity building.

Poor site selection can lead to delay in completion, rise in cost, protocol amendment, implementation variations or study failure.[24] Studies have used different site selection processes: active hunting, peer referrals, inviting expression of interests, engaging research organisations, etc.[25–27] Our organisation has been conducting multisite network studies (ranging from 5 to 84 partner institutions) over last 15 years. We have adopted variable combination of

processes for the study site and investigators selection, which have evolved over time. For this study, we adopted a systematic site selection process, which appeared lengthy and required several rounds of interactions. The four-step process assisted in identifying the highly motivated investigators and study sites with desired capacity. The potential investigators who failed to return the questionnaire with desired information or participate in the interaction are unlikely to devote time for study conduct and supervision. The systematic selection process allowed reducing the potential sites to a manageable number of suitable sites that were further assessed through visit. While the process was lengthy, it allowed also to build common understanding and commitment for the study. These experiences were similar to some multicounty studies.[25 27 28] The experiences from longer tenure clinical trial network (CTN) suggest that adoption of objective, standardised and systematic approach of site selection has better performance than an informal process. A CTN experimenting interventions for substance abuse moved from informal site selection process initially to five-step process including identification of potential sites, site selection surveys, pilot simulation data abstraction, blinded review, site selection interviews or site visits.[27] A CTN experimenting surgical interventions adopted five-step site selection process including open call, site capacity survey and pilot simulation data abstraction, criteria-based evaluation, telephonic interview and final assessment for selection.[25] The steps adopted were comparable to the steps adopted in our study.

The procedural efforts and time taken for ethical approval from the site institutes are well known. A review observed that the study site ethics approval took from 5 to 798 days and the review process and contents varied widely.[29] The ethics review process consumed sizeable staff hours and budget, forcing timeline extension and budget shortfall.[29–31] In our study also, the documentation and formats used for submission varied widely, apart from the study protocol, CRFs and consent forms. Some of the committees asked the investigators to present their protocol, while others did not.

Although the investigators were responsible for the conduct of the study, the agreements with institutions assisted in implementation of the study. The involvement of external experts TAG as in study site selection and monitoring facilitated standardisation and quality assurance. The additional data retrieval and verification effort at the sites documented the protocol adherence and robustness of data collection.

This is the first documentation of systematic site selection process for a multisite network in India. The positive experience from this study encouraged adoption of similar systematic site selection and data quality assurance mechanism for subsequent two vaccine network studies in India.

There are some limitations in the current study and documentation. While many of the lessons may be generic and have relevance for most of the multisite studies, some of them may be relevant for India and the developing countries. The documentation of contractual and financial management challenges may be limited. The study was not a clinical trial and did not involve any laboratory procedures. This was a descriptive study and had no comparison or control arm to compare the experience.

## CONCLUSIONS

In conclusion, our experience with this systematic study site selection process was positive and satisfying. It used a four-step selection process using questionnaire, objective criteria for assessment and interactions with the investigators and institution stakeholders with site visits to identify suitable sites with adequate capacity. The participation of motivated instigators, agreement with the institutions and contributory protocol and tool development facilitated successful completion. Difficulties and delays in site initiation were primarily due to the regulatory and administrative approvals. The multistep site selection adopted is feasible and practical even in Indian context. The quality assurance processes also assisted in high-quality data collection. We hope that the site selection and quality assurance processes would be informative for multisite studies in India and developing countries and appropriate adaptation or modifications may be needed as per the objectives and implementation protocol of the network studies.

**Acknowledgements** We acknowledge the support from Ministry of Health and Family Welfare, Government of India for undertaking the study. We are thankful to the hospital administrations and the clinicians at the study site institutes, who supported and facilitated undertaking of the study. We highly value the technical guidance and inputs provided by the members of Technical Advisory Group: Satinder Aneja, Anju Seth and Archana Puri, Female Hardinge Medical College, New Delhi; Ashok Patwari, Hamdard Institute of Medical Sciences & Research, New Delhi; Yogesh Kumar Sarin, Maulana Azad Medical College, New Delhi; Rakesh Aggarwal, Anshu Srivastava and Ujjal Poddar, Sanjay Gandhi Postgraduate Institute of Medical Sciences, Lucknow; Malathi Satyasekharan, Kanchi Kamakoti Chailds Trust Hospital, Chennai; Raju Sharma and Nirupam Madan, All India Institute of Medical Sciences, New Delhi; Jyoti Joshi and Deepak Polpakara, Immunisation Technical Support Unit; Ministry of Health & Family Welfare, New Delhi; Umesh D Parashar; Centres of Disease Control and Prevention, Atlanta, USA; Naveen Thacker, Child Health Foundation, Gandhigram; and Rashmi Arora, Indian Council of Medical Research, Ansari Nagar, New Delhi. We acknowledge the contribution of the research staffs at The INCLEN Trust International:Harshpreet Kaur, Janvi Chaubey, Mrimmaya Das, Shweta Sharma and Vaibhav Jain. We highly appreciate the efforts made by the research staffs at the study sites: Aarezo Bashir and Rafia; Sher-e-Kashmir Institute of Medical Sciences, Srinagar, Jammu & Kashmir; Prabha Shankar, Medanta-The Medicity Hospital, Gurgaon, Haryana; Anju Sharma; Maulana Azad Medical College, New Delhi; Anita Singh and Shubhranshu Srivastava, King George Medical University, Lucknow, Uttar Pradesh; Hemant Meena, Choithram Hospital, Indore, Madhya Pradesh; Pankaj Kumar and Shashi Kant; Indira Gandhi Institute of Medical Sciences, Patna, Bihar; Goutam Benia, IMS & SUM Medical College & Hospital, Bhubaneshwar, Odisha; Prasntajyoti Mohanty, SVP Post Graduate Institute of Paediatrics, Cuttack, Odisha; Angshuman Chatterjee, Institute of Postgraduate Medical Education and Research & SSKM Hospital, Kolkata, West Bengal; S Yamuna, Andhra Medical College, Vishakhapatnam, Andhra Pradesh; Srinidhi Sudan, Apollo Hospitals, Hyderabad, Telangana; Rajesh Francis, Apollo Hospitals, Chennai, Tamil Nadu; T Easter Chandru, PSG Institute of Medical Sciences, Coimbatore, Tamil Nadu; Deepthy R, Julie and Anju Shivkumar, Government Medical College & SAT Hospital, Thiruvananthapuram, Kerala; Archit Vaidya, Grant Medical College & JJ Hospital, Mumbai, Maharashtra; Nimesh Chouksey, MP Shah Government Medical College, Jamnagar, Gujarat; Nidhi Singh, Fortis Escorts Hospital, Jaipur, Rajasthan; Mrinmoy Gohain, Gauhati Medical College,

Guwahati, Assam; Arpita Bhattachrjee, Saugat Ghosh and Tanusmita Debnath, Agartala Government Medical College, Agartala, Tripura.

**Collaborators** The INCLEN Intussusception Surveillance Network Study Group: Manoja Kumar Das (Director Projects, The INCLEN Trust International, New Delhi, India). Narendra Kumar Arora (Executive Director, The INCLEN Trust International, New Delhi, India). Arindam Ray (Senior Program Officer, Bill and Melinda Gates Foundation, India Country Office, New Delhi, India). Ashish Wakhlu (Professor, Department of Paediatric Surgery, King George's Medical University, Lucknow, Uttar Pradesh, India). Bhadresh R Vyas (Professor, Department of Paediatrics, MP Shah Government Medical College, Jamnagar, Gujarat, India). Javeed Iqbal Bhat (Assistant Professor, Department of Paediatrics, Sher-I-Kashmir Institute of Medical Sciences, Srinagar, Jammu & Kashmir, India). Jayanta K Goswami (Professor, Department of Paediatric Surgery, Gauhati Medical College, Guwahati, Assam, India). John Mathai (Professor, Department of Paediatrics, PSG Institute of Medical Sciences, Coimbatore, Tamil Nadu, India). Kameswari K (Professor, Department of Paediatric Surgery, Andhra Medical College, Vishakhapatnam, Andhra Pradesh, India). Lalit Bharadia (Consultant Paediatric Gastroenterologist, Fortis Escorts Hospital, Jaipur, Rajasthan, India). Lalit Sankhe (Assistant Professor, Department of Community Medicine, Grant Medical College & JJ Hospital, Mumbai, Maharashtra, India). Ajaya Kumar MK (Professor, Department of Paediatric Surgery, Government Medical College & SAT Hospital, Thiruvananthapuram, Kerala, India). Neelam Mohan (Consultant Paediatrics Gastroenterology, Medanta—The Medicity, Gurgaon, Haryana, India). Pradeep K Jena (Professor, Department of Paediatric Surgery, SCB Medical College, Cuttack, Odisha, India). Rachita Sarangi (Professor, Department of Paediatrics, IMS & SUM Medical College & Hospital, Bhubaneswar, Odisha, India). Rashmi Shad (Consultant Paediatrics, Choithram Hospital and Research Centre, Indore, Madhya Pradesh, India). Sanjib K Debbarma (Associate Professor, Department of Paediatrics, Agartala Government Medical College, Agartala, Tripura, India). Shyamala J (Consultant Paediatrics, Apollo Hospitals, Chennai, Tamil Nadu). Simmi K Ratan (Professor, Department of Paediatric Surgery, Maulana Azad Medical College, Delhi, India). Suman Sarkar (Assistant Professor, Department of Paediatrics, Institute of Post Graduate Medical Education and Research, Kolkata, West Bengal, India). Vijayendra Kumar (Professor, Department of Paediatric Surgery, Indira Gandhi Institute of Medical Sciences, Patna, Bihar, India). Anand P Dubey (Professor, Department of Paediatrics, Maulana Azad Medical College, Delhi, India). Atul Gupta (Consultant Paediatric Surgery, Fortis Escorts Hospital, Jaipur, Rajasthan, India). Bikasha Bihary Tripathy (Associate Professor, Department of Paediatric Surgery, IMS & SUM Medical College & Hospital, Bhubaneswar, Odisha, India). Cenita J Sam (Professor, Department of Paediatric Surgery, PSG Institute of Medical Sciences, Coimbatore, Tamil Nadu, India). Gowhar Nazir Mufti (Assistant Professor, Department of Paediatric Surgery, Sher-I-Kashmir Institute of Medical Sciences, Srinagar, Jammu & Kashmir, India). Harsh Trivedi (Professor, Department of Paediatric Surgery, MP Shah Government Medical College, Jamnagar, Gujarat, India). Jimmy Shad (Consultant Paediatric Surgery, Apollo Hospitals, Chennai, Tamil Nadu, India). Kaushik Lahiri (Consultant, Department of Paediatric Surgery, Gauhati Medical College, Guwahati, Assam, India). Meera Luthra (Consultant Paediatric Surgery, Medanta- The Medicity, Gurgaon, Haryana, India). Padmalatha P (Professor, Department of Paediatrics, Andhra Medical College, Vishakhapatnam, Andhra Pradesh, India). Rakesh Kumar (Associate Professor, Department of Paediatrics, Indira Gandhi Institute of Medical Sciences, Patna, Bihar, India). Ruchirendu Sarkar (Professor, Department of Paediatric Surgery, Institute of Post Graduate Medical Education and Research, Kolkata, West Bengal, India). Santosh Kumar A (Professor, Department of Paediatric Surgery, Government Medical College & SAT Hospital, Thiruvananthapuram, Kerala, India). Subrat Kumar Sahoo (Associate Professor, Department of Paediatric Surgery, IMS & SUM Medical College & Hospital, Bhubaneswar, Odisha, India). Sunil K Ghosh (Associate Professor, Department of Pediatric Surgery, Agartala Government Medical College, Agartala, Tripura, India). Sushant Mane (Assistant Professor, Department of Paediatrics, Grant Medical College & JJ Hospital, Mumbai, Maharashtra, India). Bashir Ahmad Charoo (Professor, Department of Paediatrics, Sher-I-Kashmir Institute of Medical Sciences, Srinagar, Jammu & Kashmir, India). Rajendra Prasad G (Professor, Department of Paediatric Surgery, Andhra Medical College, Vishakhapatnam, Andhra Pradesh, India). Harish Kumar S (Paediatrics Radiologist, Apollo Hospitals, Chennai, Tamil Nadu, India). Jothilakshmi K (Professor, Department of Paediatrics, PSG Institute of Medical Sciences, Coimbatore, Tamil Nadu, India). Nihar Ranjan Sarkar (Associate Professor, Department of Radiology, Institute of Post Graduate Medical Education and Research, Kolkata, West Bengal, India). Pavai Arunachalam (Professor, Department of Paediatric Surgery, PSG Institute of Medical Sciences, Coimbatore, Tamil Nadu, India). Satya SG Mohapatra (Professor, Department of Radiology, IMS & SUM Medical College & Hospital, Bhubaneswar, Odisha, India). Saurabh Garge (Consultant Paediatric Surgery, Choithram Hospital and Research Centre, Indore, Madhya Pradesh, India).

**Contributors** Study conceptualisation, study design, protocol development, training, data analysis, interpretation, coordination, monitoring and quality assurance: MKD and NKA. Participant recruitment and data collection: AW, BRV, JIB, JKG, JM, KK, LB, LS, MAK, NM, PKJ, RS-1, RS-2, SKD, SJ, SKR, SS, VK, APD, AG, BBT, CJS, RPG, GNM, HT, SJ, KL, ML, PP, RK, RS-3, SKA, SKS, SKG, SM, BAC, GRP, SHK, KJ, NRS, PA, SSM and SG. Manuscript preparation: MKD, NKA and AR. All authors reviewed, provided critical input and approved the final version. The contents represent the views of the authors alone and do not necessarily represent the official positions of their organisations.

**Funding** This project was supported by the Bill and Melinda Gates Foundation, USA to The INCLEN Trust International (grant number OPP1116433).

**Disclaimer** The funder or its representative had no role in the design of the study and collection, analysis and interpretation of data and writing the manuscript.

**Map disclaimer** The depiction of boundaries on this map does not imply the expression of any opinion whatsoever on the part of BMJ (or any member of its group) concerning the legal status of any country, territory, jurisdiction or area or of its authorities. This map is provided without any warranty of any kind, either express or implied.

**Competing interests** None declared.

**Patient consent for publication** Not required.

**Ethics approval** The study protocol was reviewed and approved by all the participating institutes. The list of ethics committees of the participating institutes include the following: The INCLEN Independent Ethics Committee, The INCLEN Trust International, New Delhi, India (Ref No: IIEC 23; Dated 30 June 2015); Institutional Ethics Committee, King George's Medical University, Lucknow, Uttar Pradesh, India (Ref no: 7951/Ethics/R.Cell-15; Dated 4 December 2015); Institutional Ethics Committee, MP Shah Government Medical College, Jamnagar, Gujarat, India (ref no: 01/46/2016; Dated 6 January 2016); Institutional Ethics Committee, Sher-I-Kashmir Institute of Medical Sciences, Srinagar, Jammu & Kashmir, India (Ref no: 42/2015; Dated 29 October 2015); Institutional Ethics Committee, Gauhati Medical College, Guwahati, Assam, India (Ref no: MC/02/2015/274; Dated 30 May 2016); Institutional Human Ethics Committee, PSG Institute of Medical Sciences, Coimbatore, Tamil Nadu, India (Ref no 15/294; Dated 16 November 2015); Institutional Ethics Committee, King George Hospital, Andhra Medical College, Vishakhapatnam, Andhra Pradesh, India (Dated 26 October 2015); Institutional Ethics Committee, Fortis Escorts Hospital, Jaipur, Rajasthan, India (Ref no: FEHJ/IEC/15/0023; Dated 14 September 2015); Institutional Ethics Committee, Grant Medical College & JJ Hospital, Mumbai, Maharashtra, India (Ref no: IEC/Pharm/288/15; Dated 19 November 2015); Institutional Ethics Committee, Government Medical College & SAT Hospital, Thiruvananthapuram, Kerala, India (Ref no: 06/05/2015/MCT; Dated 9 December 2015); Medanta Institutional Ethics Committee, Medanta-The Medicity, Gurgaon, Haryana, India (MICR 559/2015; Dated 21 January 2016); Institutional Ethics Committee, SCB Medical College, Cuttack, Odisha, India; Institutional Ethics Committee, IMS & SUM Medical College & Hospital, Bhubaneswar, Odisha, India (Ref no 210/5/10/2015; Dated 14 October 2015); Institutional Ethics Committee; Choithram Hospital and Research Centre, Indore, Madhya Pradesh, India (Ref no: EC/Oct/15/20; Dated 27 October 2015); Institutional Ethics Committee, Agartala Government Medical College, Agartala, Tripura, India; Institutional Ethics Committee- Clinical Studies, Apollo Hospitals, Chennai, Tamil Nadu (Dated 14 October 2015); Institutional Ethics Committee, Maulana Azad Medical College, Delhi, India (Ref no: F.1/IEC/MAMC/50/4/2015/308; Dated 20 November 2015); Institutional Ethics Committee, Institute of Post Graduate Medical Education and Research, Kolkata, West Bengal, India (Ref no: Inst/IEC/2016/197; Dated 1 March 2016); Institutional Ethics Committee, Indira Gandhi Institute of Medical Sciences, Patna, Bihar, India (Ref no: 1256/Acad; Dated 11 November 2016) and Ethics Committee, Apollo Hospital, Hyderabad, Telengana, India (Dated 13 October 2015). The interviews with stakeholders were done after obtaining written informed consent.

**Provenance and peer review** Not commissioned; externally peer reviewed.

**Data availability statement** This manuscript reports the processes and experience from the multisite network research. All relevant data has been reported. All data is available with the investigators and can be provided by the corresponding author upon reasonable request.

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
