## [Reviewer comments · BMJ Open]

ARTICLE DETAILS

TITLE (PROVISIONAL)	Experience of establishing and coordinating a nationwide network for bi-directional intussusception surveillance in India: Lessons for multisite research studies
AUTHORS	Study Group, The INCLEN Intussusception Surveillance; as, Manoj Kumar

VERSION 1 – REVIEW

REVIEWER	Cho, Hye-Kyung Gachon University, Department of Pediatrics
REVIEW RETURNED	01-Feb-2021

GENERAL COMMENTS	This paper seems to be intended to suggest that applying a systematic four-step approach is helpful to select sites for multicenter study to investigate intussusception in children in developing countries such as India. However, it would be possible to determine whether the selected site is appropriate or not depending on what data of intussusception is aimed to know, but this study did not provide the purpose of the surveillance. It was said that ICD, ICD-9, and ICD-10 were used as the disease code system. If the code system for intussusception or acute abdomen was different between each center, it is necessary to describe them. And if the numbers of intussusception cases differed between each centers, there have to be description about those differences and how to adjust them.
---

REVIEWER	Russell, F. M. The UNiversity of Melbourne
REVIEW RETURNED	08-Feb-2021

GENERAL COMMENTS	The paper is about the selection process for choosing surveillance sites in India for intussusception. There are very few papers outlining the process of undertaking a multi-site surveillance safety surveillance. Hence there are important lessons to be learnt and shared for others to follow. MAJOR COMMENTS Abstract 1. There is no mention in the abstract of what the surveillance was being undertaken nor the criteria for selection. Add: from the establishment of IS surveillance lessons can be applied to the selection of sites for other surveillance activities. Methods
---

	2. Study design: the study design for the IS was retrospective and prospective studies. However this paper focuses on the lessons learnt. Please change the design to reflect the objective/purpose of the paper. MINOR COMMENTS Pg 12, Line 29/30: Start new paragraph here as the first paragraph is too long.
--	---

VERSION 1 – AUTHOR RESPONSE

Reviewer: 1

This paper seems to be intended to suggest that applying a systematic four-step approach is helpful to select sites for multicenter study to investigate intussusception in children in developing countries such as India. However, it would be possible to determine whether the selected site is appropriate or not depending on what data of intussusception is aimed to know, but this study did not provide the purpose of the surveillance. It was said that ICD, ICD-9, and ICD-10 were used as the disease code system. If the code system for intussusception or acute abdomen was different between each center, it is necessary to describe them.

And if the numbers of intussusception cases differed between each centers, there have to be description about those differences and how to adjust them.

Response: We thank the reviewer for the comments to improve the manuscript.

The objective of this surveillance network was to generate background information on intussusception epidemiology in Indian children and serve as baseline for future surveillance to identify any change after vaccine introduction and address the vaccine safety concerns. We have modified the statement. (Page 12, lines 249-253)

The ICD coding system was used at all sites except two private hospitals. These two private hospitals maintained electronic medical records according to the diagnosis of the cases. We have added this in the results section (Page 17, lines 371-372)

There was no variation in the codes for intussusception and acute abdomen conditions between the hospitals/centres, as we used the ICD-9 or ICD-10 codes for the disease conditions while reviewing and retrieving cases. The ICD-9 and ICD-10 codes along with the disease conditions used are given as Supplementary document 3. (Page 14, Line 308 and Supplementary document 3)

The numbers of intussusception cases recruited at the study sites differed, as mentioned in the Data Collection section. (Page 17, lines 358-367). We have added the data analysis considerations also, as suggested. (Page 17, lines 367-369). The findings for the retrospective and prospective surveillance components have been published already (Reference 22 and 23, mentioned in page 17 lines 360 and 364).

Reviewer: 2

The paper is about the selection process for choosing surveillance sites in India for intussusception. There are very few papers outlining the process of undertaking a multi-site surveillance safety surveillance. Hence there are important lessons to be learnt and shared for others to follow.

Response: We thank the reviewer for the comments to improve the manuscript.

MAJOR COMMENTS

Abstract

1. There is no mention in the abstract of what the surveillance was being undertaken nor the criteria for selection. Add: from the establishment of IS surveillance lessons can be applied to the selection of sites for other surveillance activities.

Response: We have added the nature of the surveillance. (Page 9, lines 200-202)

We have also added the statement in conclusion, as suggested. (Page 9-10, lines 218-221)

Methods

2. Study design: the study design for the IS was retrospective and prospective studies. However this paper focuses on the lessons learnt. Please change the design to reflect the objective/purpose of the paper.

Response: As suggested we have added the sentence: "The experiences and lessons presented here are based on the concurrent documentation of processes and retrospective review of the study documents." (Page 13, lines 268-270)

MINOR COMMENTS

3. Pg 12, Line 29/30: Start new paragraph here as the first paragraph is too long.

Response: We have broken the first paragraph into two, as suggested. (Page 12, line 247)

Additional modifications

To improve the readability and sequencing of the issues, we have moved the section "Intra- and inter-department coordination" to later. (Moved from Page 18, lines 393-400 to Page 19-20, lines 425-432)

Response to the additional Editorial comments April 1, 2021

1. Ringgold ID for the author's institution: The Ringgold ID for the author's institution is 539737.

2. Inclusion of Author GRP in the contributor statement: The author name initial was wrongly typed. The correct initials are RPG (author no 39 in the list, page no 7)

3. Inclusion of Supplementary document 3 legend: The legend has been corrected (Page no 35).

VERSION 2 – REVIEW

REVIEWER	Cho, Hye-Kyung Gachon University, Department of Pediatrics
REVIEW RETURNED	20-Apr-2021

GENERAL COMMENTS	This study is a descriptive study of a four-step approach to select the investigational sites for a nationwide research on intussusception in India. Although the authors have described the limitations of this study, the biggest limitation is that this study is not a comparative or controlled study, but a descriptive study on this approach. It does not mean that this four-step approach is effective or reliable in a network study for intussusception. Please make this more clear.
---

REVIEWER	Russell, F. M. The UNiversity of Melbourne
REVIEW RETURNED	12-Apr-2021

GENERAL COMMENTS	All my queries have been addressed
------------------------------------

VERSION 2 – AUTHOR RESPONSE

Reviewer: 1

This study is a descriptive study of a four-step approach to select the investigational sites for a nationwide research on intussusception in India.

Although the authors have described the limitations of this study, the biggest limitation is that this study is not a comparative or controlled study, but a descriptive study on this approach. It does not mean that this four-step approach is effective or reliable in a network study for intussusception. Please make this more clear.

Response: We thank the reviewer for the comments to improve the manuscript. We have made the necessary changes in the limitation section of the discussion. (Changes marked copy page 23, lines 520-521)

We have modified the conclusion section also accordingly. (Changes marked copy page 24, lines 533-535)

We have added the experiences from the clinical trial networks regarding the site selection process to indicate the similarities and differences in the discussion section. (Changes marked copy page 22, lines 490-499)

We hope the revisions are acceptable.

Reviewer: 2

All my queries have been addressed

Response: We thank the reviewer for the guidance and suggestions.

Editorial comments

1. In ScholarOne system, you have indicated 'Yes' that you have completed and uploaded the relevant checklist for your article but I can see that there's no checklist uploaded. If there's any please upload the file on ScholarOne under the file designation Research Checklist.

Response: We have uploaded the STROBE Checklist.

2. Please write a correct format of abstract for research paper (max. 300 words).

Response: We revised the Abstract as per the suggested format. (Changes marked copy pages 9-10)